# Bladder Cancer Extracellular Vesicles Elicit a CD8 T Cell-Mediated Antitumor Immunity

**DOI:** 10.3390/ijms23062904

**Published:** 2022-03-08

**Authors:** Carlos J. Ortiz-Bonilla, Taylor P. Uccello, Scott A. Gerber, Edith M. Lord, Edward M. Messing, Yi-Fen Lee

**Affiliations:** 1Department of Pathology and Laboratory Medicine, University of Rochester Medical Center, Rochester, NY 14642, USA; carlos_ortizbonilla@urmc.rochester.edu; 2Department of Immunology, Microbiology and Virology, University of Rochester Medical Center, Rochester, NY 14642, USA; taylor_uccello@urmc.rochester.edu (T.P.U.); scott_gerber@urmc.rochester.edu (S.A.G.); edith_lord@urmc.rochester.edu (E.M.L.); 3Department of Surgery, University of Rochester Medical Center, Rochester, NY 14642, USA; 4Wilmot Cancer Institute, University of Rochester Medical Center, Rochester, NY 14642, USA; edward_messing@urmc.rochester.edu; 5Department of Urology, University of Rochester Medical Center, Rochester, NY 14642, USA

**Keywords:** extracellular vesicles, immunotherapy, CD8^+^ T cell, adjuvant therapy, bladder cancer

## Abstract

Tumor-derived extracellular vesicles (TEVs) play crucial roles in mediating immune responses, as they carry and present functional MHC-peptide complexes that enable them to modulate antigen-specific CD8^+^ T-cell responses. However, the therapeutic potential and immunogenicity of TEV-based therapies against bladder cancer (BC) have not yet been tested. Here, we demonstrated that priming with immunogenic Extracellular Vesicles (EVs) derived from murine MB49 BC cells was sufficient to prevent MB49 tumor growth in mice. Importantly, antibody-mediated CD8^+^ T-cell depletion diminished the protective effect of MB49 EVs, suggesting that MB49 EVs elicit cytotoxic CD8^+^ T-cell-mediated protection against MB49 tumor growth. Such antitumor activity may be augmented by TEV-enhanced immune cell infiltration into the tumors. Interestingly, MB49 EV priming was unable to completely prevent, but significantly delayed, unrelated syngeneic murine colon MC-38 tumor growth. Cytokine array analyses revealed that MB49 EVs were enriched with pro-inflammatory factors that might contribute to increasing tumor-infiltrating immune cells in EV-primed MC-38 tumors. These results support the potential application of TEVs in personalized medicine, and open new avenues for the development of adjuvant therapies based on patient-derived EVs aimed at preventing disease progression.

## 1. Introduction

Extracellular vesicles (EVs) are small round double-membrane-enclosed vesicles secreted by most cell types that can transport a wide variety of cargos including proteins, nucleic acids and lipids [1]. EVs are classified into different subtypes based on their intracellular compartment of origin, surface protein markers and size, which can range from 30 to 1000 nm [2]. While initially disregarded as a form of cellular garbage, in the late 1990s, it was discovered that EVs could play a variety of functional roles in the physiology and pathophysiology of different diseases, including cardiovascular diseases, diabetes and cancer. Accumulating evidence indicated that tumor-derived EVs (TEVs) play an integral role in all stages of cancer progression, functioning to promote neoplastic transformation, cancer cell survival, metastasis, angiogenesis and immune suppression [3,4,5,6,7,8]. In contrast, fewer studies have explored the roles of EVs as modulators of antitumor immune responses, and this topic remains understudied in the context of bladder cancer (BC).

BC is one of the most prevalent cancers in the world and is estimated to be the fourth most common cause of new cancer-related deaths among men in the United States in 2021 [9,10]. The majority of BC patients exhibit non-muscle invasive (NMI) disease [11,12], of whom two-thirds experience tumor recurrence within five years following the transurethral resection of primary tumors [13,14]. Approximately 25–30% of BC patients exhibit muscle-invasive (MI) disease, and 50% of these MIBC patients will develop metastatic disease, which is associated with poor 5-year survival rates [15]. High tumor recurrence rates and the high morbidity in NMI BC and mortality associated with MI BC make BC one of the most expensive cancers to treat and manage, with the highest lifetime treatment cost per patient owing to the need for constant clinical visits to monitor for potential future tumor development [16,17,18]. Therefore, better therapeutic options for BC patients are needed, such as patient-specific therapies capable of reducing the high tumor recurrence.

The application of both natural and bioengineered EVs, as well as artificial EV-like nanoparticles for therapeutic delivery systems, has been a topic of growing interest due to their nano-scale size, natural origin and ability to encapsulate various biomolecules within the lipid bilayer membrane. EVs, including certain TEVs, have been proposed to serve as a cancer vaccine given that they naturally carry antigenic peptides capable of eliciting antitumor immune responses [19,20], yet this idea has not been tested in BC. In this current study, we applied a preclinical MB49 syngeneic mouse model and found that BC-derived EVs are immunoactive and can prevent autologous tumor growth through a mechanism primarily mediated by CD8^+^ T lymphocytes. Interestingly, priming with BC-derived EVs was able to delay unrelated colon tumor growth in vivo. Further analyses revealed that these BC EVs can activate DCs, thereby eliciting robust innate and adaptive antitumor immune responses in vivo, and EV cargos were found to be enriched for proteins and cytokines related to the immune response. This work is the first to our knowledge to demonstrate that BC-derived EVs can elicit an antitumor immune response against BC as well as an unrelated tumor type, highlighting new avenues for the development of personalized cancer therapies based upon patient-derived EVs to prevent disease progression. 

## 2. Results

### 2.1. MB49 Cells Release Immunostimulatory EVs That Activate Bone Marrow-Derived Dendritic Cells

The contrasting roles of TEVs have been studied in a variety of cancer types, but further study of BC-derived EVs is warranted. To that end, the immunological properties of EVs derived from MB49 cells, which are among the most-used murine BC cell lines, were characterized. MB49 cell-derived EVs (MB49 EVs) were purified from cell culture supernatants by serial ultracentrifugation and characterized by nanoparticle tracking analysis (NTA). These MB49 EVs were within the expected size distribution for EVs, with a modal average diameter of 72.5 nm, and an average release of 43 particles/h/cell (Figure 1A). The EV cargo proteins were analyzed by Mass Spectrometry, leading to the identification of 2554 distinct proteins with more than 100 functional clusters based on PANTHER pathway analyses. Among these, around 9% of proteins were functionally associated with various immune pathways, including inflammation, cytokine production, toll-like receptor signaling and lymphocyte activation (Figure 1B, Appendix A). Western blotting analyses of purified EVs further detected the presence of key immunomodulatory proteins, including MHC-I, MHC-II and the co-stimulatory molecules CD80 and CD86 (Figure 1D), as well as the EV markers, Alix and TGS101 (Figure 1C). We then tested the ability of EVs to activate bone marrow-derived dendritic cells (BMDCs) ex vivo. MB49 EV treatment directly activated these naïve BMDCs, as evidenced by an increase in the percentage of MHC-II^+^/CD86^+^ cells relative to that observed for PBS-treated control cells (Figure 1E).

In summary, these data indicate that MB49 EVs are enriched with several cargo proteins with immunomodulatory potential and can activate BMDCs. 

### 2.2. MB49 EV Priming Protects Mice against Subsequent Autologous Tumor Growth

In light of the immune-related cargo proteins identified within MB49 EVs (Figure 1), it is possible that these EVs can provoke antitumor immunity. To explore this possibility, we designed an in vivo experiment wherein C57BL/6 mice were primed twice with MB49 EVs at 21 and 14 days prior to the subcutaneous implantation of MB49 tumor cells (Figure 2A). Surprisingly, we found that MB49 EV priming nearly completely suppressed MB49 tumor growth in vivo when compared to PBS-primed control mice (Figure 2B). Tumor tissues from these animals were harvested at two time points, weighed, and subjected to flow cytometry analyses. Consistent with the observed reduction in tumor volume over time, a significant tumor weight reduction was observed in the MB49 EV-primed group (Figure 2C). These data were also supported by a decrease in the percentage of the CD45^−^ population (e.g., tumor cells) in tumor tissues collected from MB49 EV-primed mice as detected by flow cytometry (Figure 2D).

### 2.3. MB49 EVs Modulate Intratumoral Immune Cells

The tumor microenvironment plays a decisive role in shaping tumor development [21]. To examine alterations in tumor-infiltrating innate and adaptive immune cell populations induced by MB49 EV priming, tumors/tissues collected on days 11 and 19 after implantation were analyzed by flow cytometry. While there were few differences in intratumoral immune subsets present in PBS control and EV-treated tumors on day 11, more apparent differences were observed on day 19 (Figure 3A). Specifically, we found that EV priming resulted in a significant decrease in intratumoral inflammatory monocyte and granulocyte abundance, along with significant increases in all other analyzed immune subsets (TAMs, NK, DCs, B cells, and T cells) (Figure 3B). Notably, on day 19, the intratumoral CD8^+^ T cells from MB49 EV-primed mice exhibited a significant increase in the cell surface expression of the activation marker CD69 as measured by its mean fluorescence intensity (MFI) relative to PBS-treated control mice (Figure 3C).

Taken together, these data confirmed the immunogenic potential of MB49 EVs, which functioned as a de facto therapeutic vaccine capable of protecting against subsequent MB49 tumor growth by increasing the intratumoral infiltration of different immune cell populations and enhancing the activation of cytotoxic T lymphocytes in vivo.

### 2.4. CD8^+^ T Cells Mediate the Antitumor Immune Response against MB49 Cells in EV-Primed Mice

Cytotoxic CD8^+^ T cells very often mediate antitumor immune responses. We hypothesized that BC EV-induced antitumor immune response is largely attributable to the observed enhanced infiltration and activation of CD8^+^ T lymphocytes shown in Figure 3. To test this hypothesis, CD8^+^ T lymphocytes were depleted in vivo by the intraperitoneal administration of an anti-CD8 antibody, starting from one day prior to tumor cell injection until day 17 [22,23] (Figure 4A). As expected, MB49 EV priming alone elicited antitumor activity such that MB49 tumors were not measurable 25 days post tumor implantation (Figure 4B), consistent with complete tumor rejection as it was later confirmed by dissection and gross histology (Figure 4C). Aligned with our hypothesis, administration of neutralizing CD8^+^ T lymphocytes in vivo completely abrogated the antitumor effects of MB49 EV priming such that no differences in tumor growth were observed among the groups (Figure 4B). Moreover, the increased tumor size observed in groups of mice in which CD8^+^ T lymphocytes were neutralized (black and gray dotted lines) irrespective of MB49 EV priming status underscores CD8^+^ T cells’ role in controlling tumor growth even in control PBS-primed mice (solid black line). These data suggest that CD8^+^ T lymphocytes are likely mediating an antigen-specific immune response induced by MB49 EVs.

### 2.5. MB49 EVs Enhance Innate Immune Cell Tumor Infiltration and Delay Tumor Growth in an Unrelated Syngeneic Cancer Model

EVs have been shown to carry MHC antigen complexes capable of eliciting an antigen-specific immune response [24]. As we observed that MB49 EV priming was sufficient to abolish the establishment of MB49 tumors, we hypothesized that this observation may be attributable to the induction of an MB49 cell-specific immune response. To test this hypothesis, we examined the protective effect of MB49 EVs against unrelated tumors in C57BL/6 mice by priming these animals with MB49 EVs and later injecting them with syngeneic murine Colon.38 cancer cells derived from the same B6 genetic background (Figure 5A). Interestingly, we observed reduced Colon.38 tumor growth in MB49 EV-primed animals as compared to PBS-primed control animals (Figure 5B). However, the level of the antitumor immune response was less pronounced relative to that observed for autologous MB49 tumors (Figure 2, Figure 3 and Figure 4), wherein complete tumor rejection was achieved. We analyzed Colon.38 tumor-infiltrating immune cell populations on day 19 and detected no significant increase in CD8^+^ T lymphocyte levels (Figure 5D), as seen in the autologous tumor model (Figure 4). Nonetheless, we detected significant increases in tumor-infiltrating inflammatory monocytes, DCs, and CD4^+^ T cells in MB49 EV-primed animals (Figure 5D). The percentages of infiltrating granulocytes also trended towards an increase following MB49 EV priming, although the difference was not statistically significant. These findings suggest that MB49 EVs may provoke an immune response against cancer cells in part via the delivery of other immunomodulatory factors, such as cytokines, thereby affecting the host immune response and/or directly inducing tumor cell death. To explore this possibility, a cytokine array analysis was performed to detect the presence of 96 different pro-inflammatory cytokines in MB49 EVs (Appendix A, Figure 5E). The results revealed the presence of several key cytokines, including eotaxin-2, MIP-2, and MCP-1, which are known to induce the recruitment of different subsets of innate immune cells (Appendix A) [25,26,27]. The presence of these cytokines in MB49 EVs highlights a potential mechanism of action underlying the observed anti-Colon.38 tumor activity, where EV-induced immune surveillance could play a role.

In summary, we found that MB49 EVs, in addition to enhancing CD8^+^ T-cell-mediated antitumor immune responses (Figure 2, Figure 3 and Figure 4), can also boost immune system activation at a more general level and promote overall immune surveillance.

## 3. Discussion

EVs are released during normal cellular activity by most cell types, including immune and tumor cells, and can mediate cell–cell communication resulting in changes in the behavior of recipient cells. TEVs contain oncogenic cargos that can be transferred to various cells within the TME, consequently promoting the development of an immunosuppressive microenvironment that is beneficial for cancer cell survival. Our study, in contrast, revealed that TEVs carry immune complexes that, rather than being immunosuppressive, are immunostimulatory and can protect mice against cancer progression. Our study reinforces the multifaceted roles of TEVs as mediators of tumor growth.

High tumor mutation burden often renders tumors more immunogenic and more responsive to immunotherapeutic intervention. BC is one of the most mutated cancers [37]; therefore, BC-derived EVs may represent a good source of tumor antigens, enabling the immune system to better recognize and eliminate these tumors. The most commonly studied immunotherapeutic approaches to treat BC include immune checkpoint blockade efforts using monoclonal antibodies specific for PD-L1, PD-1 and CTLA-4 [38]. Despite being used in patients with different types of cancer, including BC patients, improvements in patient outcomes are not always observed [38] and there are no validated biomarkers for predicting treatment responses in BC at present. These immunotherapies operate by overcoming the “misuse” of the immune checkpoints by cancer cells to escape the immune system, rather than by directly enhancing antigen-specific immune memory as a means of preventing future tumor developments. Therefore, the antitumor immunity induced by BC-derived EVs may offer an alternative therapeutic option to prevent future BC tumor recurrence and/or to complement existing therapies by providing some degree of immunospecificity. In addition, these BC-derived EVs are nano-sized, of natural origin and most importantly, they contain endogenous components (proteins and cytokines) that can be transferred into target immune cells with high efficiency. They can also be potentially used as a safe cargo-delivery agent for other desired components that could enhance antitumor immune responses. The nano-scale sizing of these EVs is an important consideration in therapeutic contexts, enabling them to deliver cargo molecules not only locally, but also throughout the body [39]. In addition, EVs are capable of crossing biological barriers, such as the blood–brain barrier, blood–lymph barrier, and placental barrier, which remains a challenge for many other treatments [40].

Our data indicated that BC cells release immunoactive EVs that can activate DCs and priming mice with BC-derived EVs was sufficient to protect them against subsequent autologous tumor challenge. These findings highlight a novel opportunity for the development of personalized medicine, and opens a door for the further investigation of the application of BC-derived EVs as adjuvant therapy in combination with immune checkpoint inhibitors and/or BCG immunotherapy as a means of enhancing antitumor immune responses and preventing the development of future BC tumors. In addition, administering autologous patient-derived EVs can also reduce the odds of adverse reactions following EV priming, given that the need for donor matching is obviated. However, it is also important to recognize that these beneficial antitumor effects of BC EVs were observed only when mice were primed prior to tumor cell injection, and the preventive effects disappear in mice with established tumors (Ortiz-Bonilla, C.J. et al., manuscript in preparation). This suggests that the timing of immunization is critical to the antitumor efficacy of these EVs [41]. One possible explanation is that when cancer cells escape from immune surveillance, the newly established tumors may have decreased cell surface MHC expression and/or experience antigen loss, thus enabling them to evade EV-mediated immune recognition. Moreover, the TME in tumor-bearing mice may be more conducive to sustained tumor growth [42], which could negatively influence the effector functions of infiltrated immune cells. To note, if the EV priming takes place after tumor establishment, those tumors might have already evolved and escaped from immune surveillance, leading to an unsuccessful antitumor immune response. Therefore, identifying the intrinsic and environmental factors that contribute to these phenotypes has the potential to facilitate the further optimization of the ability of EVs to induce antitumor immunity.

The fact that EV priming increased the levels of tumor-infiltrating CD8^+^ T lymphocytes in EV-primed animals while the depletion of CD8^+^ T lymphocytes completely abrogated the beneficial impact of EV treatment suggests that these TEVs might provoke an antigen-specific T-cell response. This is not surprising, since these lymphocytes are well known to be involved in antitumorigenic action in several cancer models, with supporting evidence from clinical samples [43]. Note that the inhibition of CD8^+^ T cells elevated MB49 tumor burden even in the PBS-primed animals as compared to the IgG control group. This aligns with evidence that CD8^+^ T cells are the primary mediators of antitumor immunity. However, several outstanding questions remain to be answered and warrant further study. First, the underlying mechanisms whereby BC cells release immunogenic EVs that are capable of inducing CD8^+^ T cells intratumor infiltration, whether EVs can directly promote T-cell proliferation, migration, or indirectly function as chemoattractant for T-cell infiltration remains to be determined. Second, the impact of EVs on other immunoactive tissues, such as spleen and lymph nodes, might provide additional functional mechanisms mediated by TEVs. Third, the deep dive into immunophenotyping circulating immune cells upon EV priming would also shed some light as to how these TEVs regulate various immune cells, such as NK, B, CD4^+^ and CD8^+^ T cells, to provoke an antitumor immunity.

We tested the unrelated syngeneic Colon.38 tumor model in MB49 EV-primed animals, given that mouse Colon.38 cells do not share any known tumor antigen with MB49 cells. The partial antitumor effect observed in this unrelated syngeneic tumor model may thus be a result of a non-specific innate immune response. Cytokines released by innate immune cells play a key role in regulating the immune system, and EVs have been shown to carry biologically active cytokines [44]. Among the cytokines, we detected within MB49 EVs were eotaxin-2 and MIP-2, which are known to induce the recruitment of innate immune cells such as eosinophils, basophils, neutrophils and macrophages [23,24], and MCP-1, a chemokine involved in monocyte recruitment [25,45]. Interestingly, high levels of monocyte infiltration were observed in the unrelated syngeneic tumor model. These three highlighted cytokines have been previously reported to be encapsulated in EVs from skin cancer cells [46], supporting the potential for the therapeutic application of TEVs to be generalized among other cancer types. However, whether TEV-contained cytokines have a direct effect in promoting immune surveillance, such as increasing total circulating monocytes and/or their infiltration into unrelated tumors, still requires further investigation. Overall, we demonstrated that TEV priming completely prevented tumor growth in an autologous mouse model through a process wherein CD8^+^ T lymphocytes were essential. In addition, BC-derived EVs also delayed tumor growth in an unrelated syngeneic tumor model, potentially as a result of the activation of the innate immune system by pro-inflammatory cytokines carried by BC-derived EVs and/or the indirect activation of the adaptive immune system.

Currently, there are multiple tumor vaccines available, including Sipuleucel-T, Talomigene, BCG, and vaccines for Human Papilloma Virus (HPV) and Hepatitis B Virus (HBV). HPV vaccines are classified as cancer vaccines because they protect against viruses that can lead to HPV-related cancers [47]. Other cancer vaccines, such as the HBV vaccine, can substantially reduce the incidence of liver cancer, but an incomplete HBV vaccination is associated with a high risk of liver cancer [48]. Sipuleucel-T is the first immunotherapeutic vaccine for castration-resistant prostate cancer, and only patients with advanced and metastatic prostate cancer are eligible for such treatment at present [49]. The BCG vaccine has recently been evaluated as a potential tool for enhancing the efficacy of BC-targeted BCG immunotherapy, given that this is one of the most successful immunotherapies in oncology to date. The subcutaneous vaccination of BC patients with BCG prior to intravesical BCG immunotherapy has yielded promising results in clinical trials [50]. Despite the promising activity of many cancer vaccines in such trials, however, there remains a strong need for a BC vaccine or adjuvant therapies with tumor specificity and fewer side effects. Our study presents a novel therapeutic tumor vaccine model wherein EVs derived from patient tumors may have the potential to prevent future tumor recurrence. However, more investigation to support this hypothesis is needed and the future comprehensive functional characterization of the BC-derived EV cargos with both pro- and antitumor properties will guide the development of a BC-derived EV-based cancer vaccine for the prevention of tumor recurrence.

This is the first study to our knowledge to show that BC-derived EVs can elicit an antitumor immune response against autologous and unrelated tumors. The underlying mechanism of such action starts with the release of immunogenic TEVs, which were internalized and processed by DCs. The EV-activated DCs would activate tumor-suppressive immune cells, including, but not limited to, CD8^+^ T cells, to then provoke an antitumor immune response. Overall, this work provides a valuable platform that will guide future research efforts to better understand the potential of BC EVs as a patient-specific therapeutic cancer vaccine.

## 4. Materials and Methods

### 4.1. Cell Culture

The MB49 BC cell line was obtained from the American Type Culture Collection (Manassas, VA, USA) and maintained according to their instructions in a humidified 37 °C 5% CO_2_ incubator using Dulbecco’s Modified Eagle’s Medium containing 10% fetal bovine serum (FBS; Thermo Fisher Scientific, Waltham, MA, USA) and 1% penicillin/streptomycin (REF 15140-122, Gibco^®^). The MC.38 colon cancer cell line was obtained as a gift from Dr. Scott A. Gerber’s laboratory and maintained in a humidified 37 °C 5% CO_2_ incubator using Roswell Park Memorial medium (RPMI) containing 10% fetal bovine serum (FBS; Thermo Fisher Scientific) and 1% penicillin/streptomycin (REF 15140-122, Gibco^®^).

### 4.2. EV Isolation and Nanoparticle Tracking Analysis

For MB49 EV collection, the cells were cultured in medium containing EV-depleted FBS as described previously [51]. Cell culture supernatants were processed by serial centrifugation at 400× *g* for 10 min and 10,000× *g* for 30 min to remove cells and debris and then stored at −80 °C. MB49 EVs were then isolated from thawed samples by ultracentrifugation at 100,000× *g* for 2 h at 4 °C, washed in a large volume of DPBS, and ultracentrifuged again at 100,000× *g* for 2 h at 4 °C. The resulting pellets were resuspended in a small volume of DPBS and aggregates were removed from the samples by centrifugation at 10,000× *g* for 5 min. Final total protein concentrations in these MB49 EV samples were measured via Micro BCA assay (catalog No. 23235, Thermo Fisher Scientific) and samples were stored at −80 °C until use. Particle size distributions and concentrations in EV isolates were estimated using a NanoSight NS300 instrument (Malvern Instruments, Malvern, UK). Samples were diluted 1:1000 in DPBS with negligible background signal and recorded in five video files (30 s each). The mode was identified, and the total particle number released per cell per hour was calculated.

### 4.3. Western Blotting Analysis

Whole-cell lysate and EV protein concentrations were measured using a Micro BCA Protein Assay Kit (catalog No. 23235, Thermo Fisher Scientific) and 10–15 μg protein samples were loaded into 10–12% SDS-PAGE gels for protein separation and transferred onto a polyvinylidene fluoride membrane. Membranes were stained with the following primary antibodies: B7-1 (1:300, H-208, sc-9091, rabbit polyclonal IgG, Santa Cruz Biotechnology, Dallas, TX, USA), B7-2 (1:300, BU63, sc-19617, mouse monoclonal IgG_1_, Santa Cruz Biotechnology), MHC Class I (1:700, H-300, sc25619, rabbit polyclonal IgG, Santa Cruz Biotechnology), MHC Class II I-A/I-E (1:200, M5/114.15.2, 14-5321-81, monoclonal IgG2b, eBioscience^TM^) and the EV markers: PDCD6IP-Alix (1:300, 12422-1-AP, polyclonal IgG, PROTEINTECH^®^ GROUP), TGS101 (1:000, 201-280, mAb, HOOOO7251-MO1, Abnova, Taipei City, Taiwan) and beta-actin (1:10,000, C4, sc-47778, mouse monoclonal IgG1, Santa Cruz Biotechnology). Secondary antibody staining was performed using Donkey anti-rabbit IgG-HRP (1:5000, sc-2313, Santa Cruz Biotechnology) and Goat anti-mouse IgG-HRP (1:4000, sc-2005, Santa Cruz Biotechnology).

### 4.4. Mass Spectrometry Analysis of EVs

MB49 EVs were isolated as described above and resuspended at 1 μg/μL in PBS. Mass spectrometry analyses were performed by the URMC Mass Spectrometry Resource Facility. A total of 2554 proteins were identified. A PANTHER database analysis was performed using the Immune System Pathway Analysis tool, with 137 proteins being identified as positive hits associated with immune system pathways. The full list of the identified proteins is compiled in Appendix A.

### 4.5. Bone Marrow-Derived Dendritic Cell Differentiation, Treatment, and Analysis

Bone marrow cells were harvested from the femurs of C57BL/6 mice and cultured in a humidified 5% CO_2_ incubator at 37 °C and in RPMI containing 10% FBS (Thermo Fisher Scientific), 1% penicillin/streptomycin (REF 15140-122, Gibco^®^), 20 ng/ml recombinant mouse GM-CSF (Cat number 415-ML, R&D systems, Minneapolis, MN, USA), and 5 ng/mL recombinant mouse IL-4 (Cta number 404-ML, R&D systems). On days three and five, fresh media was added, and on day seven the loosely attached cells were removed, counted, and seeded at the desired density. One day later, cells were treated with PBS, 1 × 10^6^ CFU BCG (as a positive control), or 100 ug/mL MB49-EVs and cultured for 72 h. Cells were then removed, washed, counted, and stained for flow cytometric analysis with the following antibodies: anti-mouse CD45 Ab (PE/Cyanine7, 30_F11, cat number 561868, BD Biosciences, Mississauga, ON, Canada), anti-mouse CD11c Ab (Brilliant Violet 605^TM^, n418, cat number 117333, BioLegend, San Diego, CA, USA), anti-mouse MHC Class II I-A/I-E Ab (Alexa Fluor^®^ 700, M5/114.15.2, cat number 107621, BioLegend) and anti-mouse CD86 Ab (APC/Cyanine7, GL-1, cat number 105029, BioLegend). Bone marrow-derived DCs were gated as CD45^+^, CD11c^+^ cells, and their activation levels were determined based upon their MHC Class II and CD86 surface protein levels.

### 4.6. In Vivo EV Priming, Tumor Inoculation, and Flow Cytometric Analysis

A general tumor protocol was established where mice received priming subcutaneous (s.c.) injections of PBS or 20 μg MB49-EVs on their flanks 21 and 14 days prior to the s.c. implantation of 2 × 10^5^ tumor cells (MB49 or MC38, as indicated) in the backs of female C57BL/6J. Tumor growth was then monitored and mice were sacrificed at two different time points (day 11 and day 19) to harvest tumors. Tumor weight was measured and tumors were processed into single-cell suspensions. A total of 1 × 10^6^ tumor cells were blocked with Fc Block (clone 2.4G2) and stained for 30 min at 4 °C in the dark with the following cocktail of directly conjugated primary antibodies: anti-mouse Ly6C (PE/Cyanine7, AL-21, cat number 338036, BD Biosciences), anti-mouse Ly6G (BV605, 1A8, cat number 563005, BD Biosciences), anti-mouse CD8 (PE/Cy5, 53.6.7, cat number 553034, eBiosciences), anti-mouse CD45 (PerCP/Cy5.5, 30-F11, cat number 561870, BD Biosciences), anti-mouse CD4 (APC/Cy7, GK1.5, cat number 561830, BD PharMingen), anti-mouse NK-1.1 (PE-CF594, PK136, cat number 562864, BD Biosciences), anti-mouse CD11b (eFluor 450, M1/70, cat number 48-0112-82, Invitrogen, Waltham, MA, USA), anti-mouse F4/80 (APC, BM8, cat number 17-4801-82, eBioscience), anti-mouse CD19 (BV510, 1D3, cat number 562956, BD Biosciences) and anti-mouse CD69 (PE, H1.2F3, cat number 553237, BD Biosciences). All samples were then washed with 1 mL of PBS 1% BSA 0.1% azide (PAB), fixed for 20 min at 4 °C in the dark with BD Cytofix/Cytoperm (BD Biosciences), washed in PAB again, and resuspended in 100 μL of PAB for analysis using a 12-color LSRII instrument (BD Biosciences) and FlowJo software (Tree Star, Ashland, OR, USA). Immune cell subpopulations were gated as indicated: granulocytes (CD45^+^, CD11b^+^, Ly6C high, and Ly6G high); inflammatory monocytes (CD45^+^, CD11b^+^, Ly6C medium, and Ly6G low); TAM (CD45^+^, CD11b^+^, Ly6C low, Ly6G low, F4/80^+^); NK cells (CD45^+^, NK1.1^+^); DCs (CD45^+^, CD11c^+^); B cells (CD45^+^, CD11b^−^, CD19^+^); CD4^+^ T cells (CD45^+^, CD4^+^); CD8^+^ T cells (CD45^+^, CD8^+^). Data are reported as a percent of CD45^+^ events recorded.

### 4.7. In Vivo CD8^+^ T-Cell Depletion

Depletion of CD8^+^ T cells in vivo was performed via the i.p. injection of 200 μg of the following antibodies: anti-mouse CD8α Ab (monoclonal, clone 53-6.7, cat number BE0004-1, Bio X Cell, Lebanon, NH, USA) or IgG2a isotype control (monoclonal, anti-trinitrophenol clone 2A3, cat number BE0089, Bio X Cell). Mice were primed twice with PBS or EVs at days 21 and 14 before injection with 2 × 10^5^ MB49 cells (s.c. in the flank). Antibody i.p. injections were administered one day prior to and one day after MB49 cell injection, and then every three days until seven injections had been completed. Tumor growth was monitored until day 25 when tumors were harvested and tumor weights were measured.

### 4.8. Pro-Inflammatory Cytokine Array

A mouse pro-inflammatory cytokine array was performed on MB49 EVs as per the manufacturer’s instructions (RayBio^®^ C-Series Mouse Cytokine Antibody Array C100, cat number AAM-CYT-1000, RayBiotech, Peachtree Corners, GA, USA). Briefly, 40 μg of EVs in 100 μL of PBS were incubated in a final concentration of 1% Triton X-100 for 1 h at room temperature with occasional vortexing to release cytokines from EVs. Then, 800 μL of PBS and 100 μL of 10X protease inhibitor (Pierce Protease Inhibitor mini tablets A32953) were added to the sample for a final sample volume of 1 mL. Samples were then added to each blocked array membrane and incubated overnight at 4 °C with gentle rocking. Membranes were washed according to the manufacturers’ instructions, and a Biotinylated Ab Cocktail was added for a 2-h incubation at room temperature. Membranes were washed again and incubated with HRP-Streptavidin for 2 h at room temperature, washed a third time, and the image analyses were then performed using ImageJ software. Background subtraction and sample normalization were performed as previously described [8] using the Microsoft^®^ Excel-based Analysis Software Tool offered by RayBiotech.

### 4.9. Statistical Analysis

Statistical analysis was performed using Prism 8 software (GraphPad, San Diego, CA, USA). Data are presented as mean § standard deviation. For paired comparisons, significance was determined by Independent Student *t*-test. For multiple group comparisons, significance was determined by two-way ANOVA with Tukey’s multiple comparisons test.

## Figures and Tables

**Figure 1 ijms-23-02904-f001:**
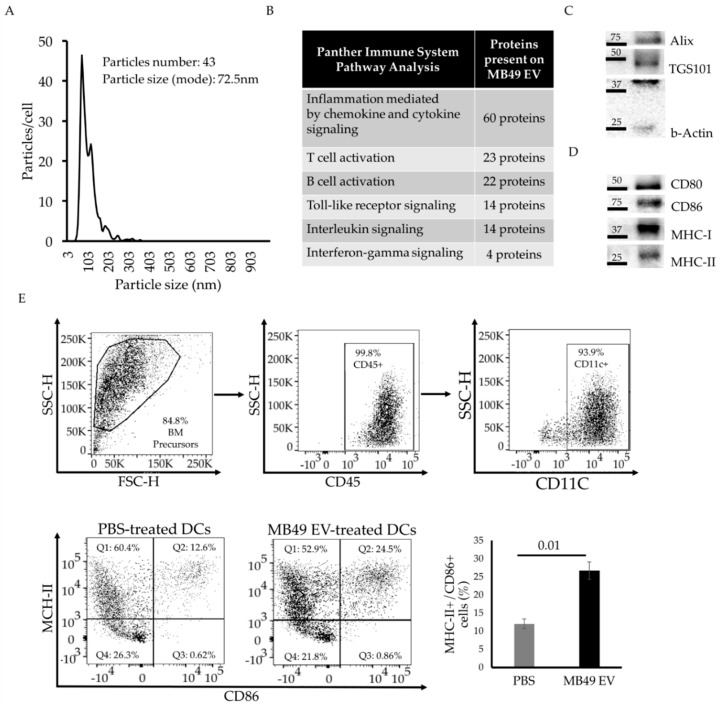
MB49 EVs contain key immunoregulatory cargo proteins and activate BMDCs. (**A**) Historam graph of number of released MB49 EVs per cell and their size (nm) estimated by Nano-particle Tracking Analysis. (**B**) Graph for the identified pathway hits using Panther database of MB49 EV protein cargo identified by Mass Spectrometry Analyses. Table listing identified immune system-related pathways and the number of proteins present on MB49 EVs for each of them. (**C**,**D**) Western blotting analysis of MB49 EVs targeting common EV markers and validating immune-related cargo proteins presence. (**E**) Gating strategy to show purity level of Bone Marrow-derived DCs in vitro differentiation and activation level changes (MHC-II and CD86) after MB49-EV treatment in vitro; quantification shown in bar graph. Error bars show standard deviation. Statistical significance was calculated by paired independent Student *t*-test.

**Figure 2 ijms-23-02904-f002:**
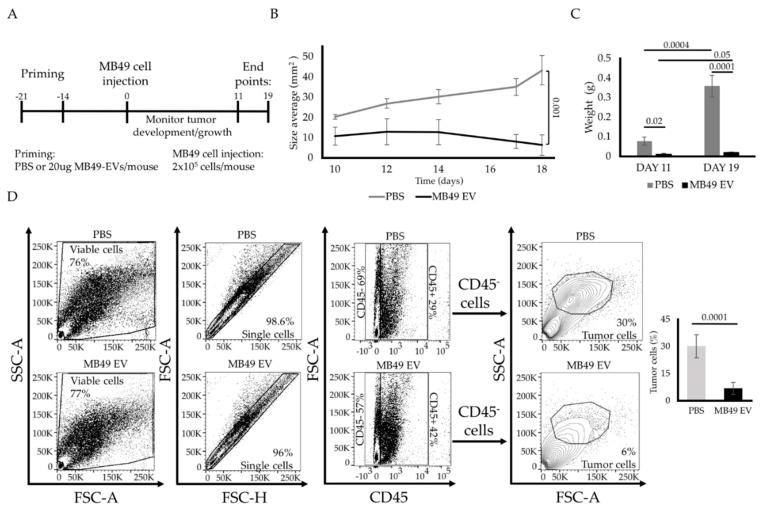
MB49 EV priming prevents MB49 tumor establishment in a syngeneic mouse model. (**A**) Diagram of in vivo experimental design timeline. (**B**) Subcutaneous MB49 tumor growth curve (mm^2^). (**C**) Bar graphs of the averaged harvested tissue weight at day 11 (left panel) and day 19 (right panel). (**D**) Flow cytometry gating strategy and percentage of tumor cells identified by cell size and complexity in harvested tissues at day 19. The *n* number is 7 on each group. Error bars show standard error. Statistical significance was calculated by independent Student *t*-test and two-way ANOVA with Tukey’s multiple comparisons test.

**Figure 3 ijms-23-02904-f003:**
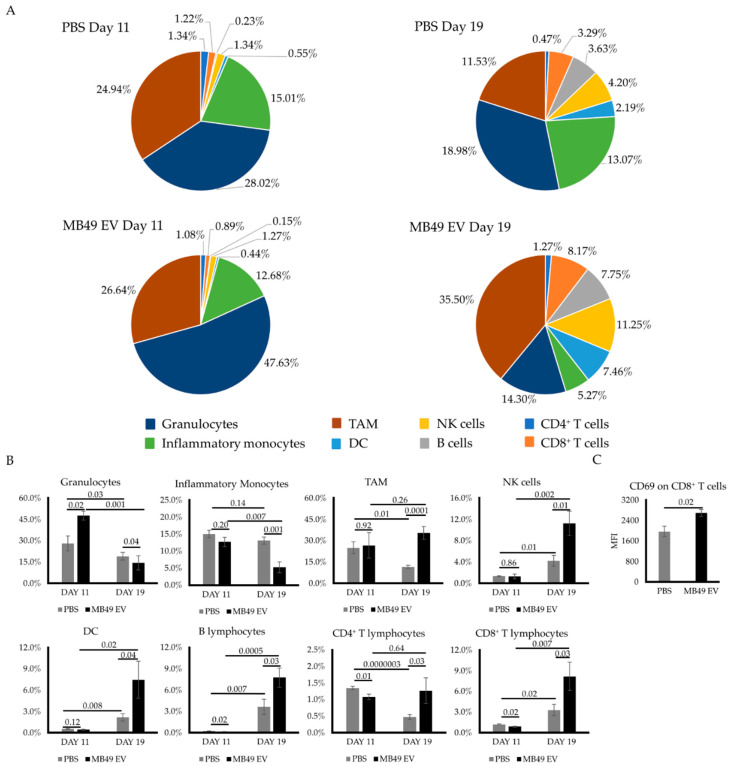
MB49 EV-driven tumor growth prevention is mediated by an immune cell-enriched TME. (**A**) Pie charts represent the percentage of tumor-infiltrated immune cell subpopulations at different time points in control PBS- and EV-primed tumors. (**B**) Bar graphs comparing averaged percentages of infiltrated immune cell subpopulations at day 11 and day 19 in control mice and MB49-EV-primed mice. Identified immune cells are shown as follows: granulocytes, inflammatory monocytes, tumor-associated macrophages, natural killer cells, dendritic cells, B lymphocytes, CD4^+^ T lymphocytes and CD8^+^ T lymphocytes. (**C**) Activation level of infiltrated CD8^+^ T lymphocytes as indicated by their surface CD69 protein expression (mean fluorescence intensity) at day 19. The *n* number is 7 on each group. Error bars show standard error. Statistical significance was calculated by independent Student *t* test and two-way ANOVA with Tukey’s multiple comparisons test.

**Figure 4 ijms-23-02904-f004:**
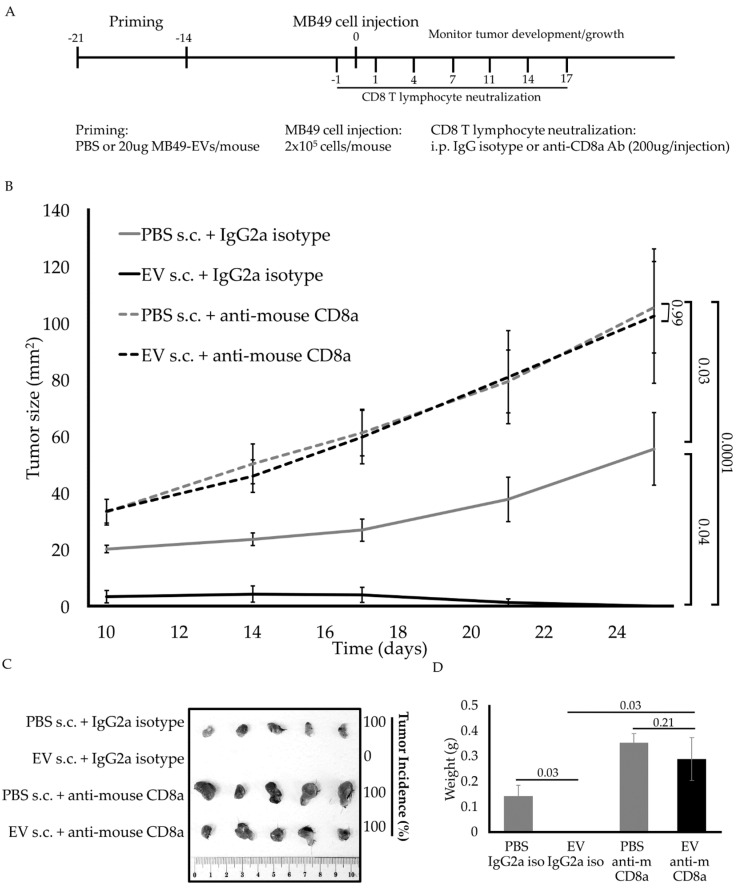
CD8^+^ T cells mediate the MB49 EV-driven antitumor immune response. (**A**) Diagram of in vivo experimental design timeline. (**B**) Subcutaneous MB49 tumor growth curve (mm^2^) showing the average tumor size for each treatment group. (**C**) Images of harvested MB49 tumor tissue after 25 days of in vivo growth. Tumor incidence percentages are shown on the right. (**D**) Bar graph showing the averaged tumor weight for each treatment group at day 25. The *n* number is 5 on each group. Error bars show standard error. Statistical significance was calculated by two-way ANOVA with Tukey’s multiple comparisons test.

**Figure 5 ijms-23-02904-f005:**
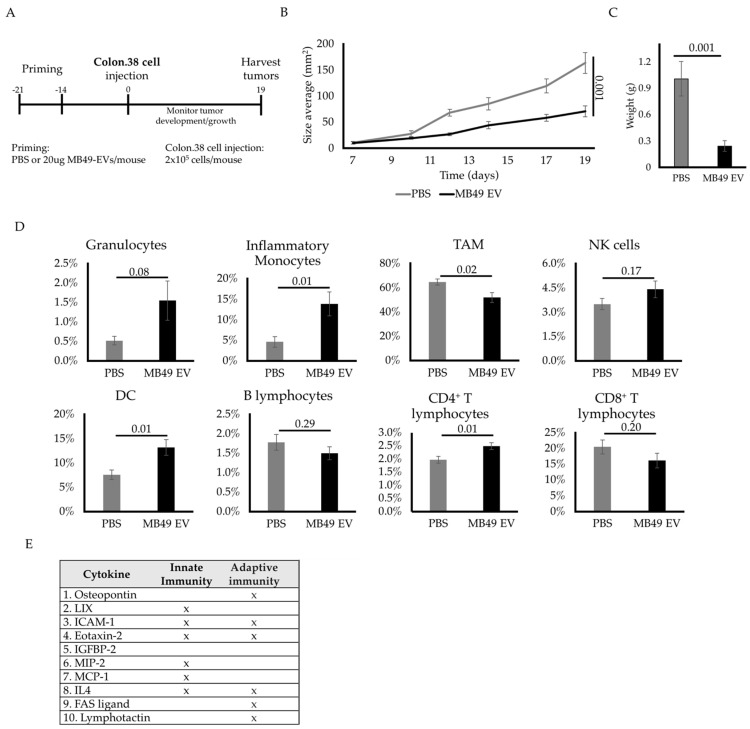
MB49 EV priming promotes both innate and adaptive immune cell-mediated antitumor immune responses. (**A**) Diagram of in vivo experimental design timeline. (**B**) Subcutaneous Colon.38 tumor growth curve after MB49-EV priming (mm^2^). (**C**) Bar graphs of the averaged harvested tissue weight at day 19. (**D**) Bar graphs comparing averaged percentages of infiltrated immune cell subpopulations at day 19 in control mice and MB49-EV-primed mice as measured by flow cytometric analysis. Identified immune cells are shown as follows: granulocytes, inflammatory monocytes, tumor-associated macrophages, natural killer cells, dendritic cells, B lymphocytes, CD4^+^ T lymphocytes and CD8^+^ T lymphocytes. The *n* number is 8 on each group. Error bars show standard error. Statistical significance was calculated by independent Student *t*-test comparing the respective two groups at the time. (**E**) Identified MB49-EV-contained cytokines and their roles in innate and adaptive immunity: osteopontin [28], LIX [29], ICAM-1 [30], Eotaxin-2 [31], IGFBP-2 [32], MIP-2 [33], MCP-1 [27], IL4 [34], PAS Ligand [35] and Lymphotactin [36].

## Data Availability

The data supporting the findings of this study are available within the articles and its Appendix A.

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
