# Peer review of "Bladder Cancer Extracellular Vesicles Elicit a CD8 T Cell-Mediated Antitumor Immunity"

_ijms, 2022, doi:10.3390/ijms23062904_

Round 1

Reviewer 1 Report

Carlos J. Ortiz-Bonilla and colleagues demonstrate that extracellular vesicles (EVs) derived from the bladder cancer cell line MB49 are characterized by several immunostimulatory proteins and are capable to activate BMDCs. Priming of mice with EVs induced a CD8 T cell response preventing the growth of autologous tumors. Additionally, BC EV priming slowed down the growth of an unrelated tumor type demonstrating the immune-stimulating character of EVs. Based on their observations, the authors conclude that BC EVs could be used as personalized cancer adjuvant therapy to prevent tumor recurrence.  

MAJOR COMMENTS/CHANGES

  1. Quality of the figures should be improved (axis labeling and scaling are blurry).
  2. Figure 2D: Why is the percentage of tumor cells in the tumor tissue so low (even in the PBS group)?
  3. Figure 3: In addition to tumor tissue, spleen, blood and/or lymph nodes should be analyzed to further characterize the immune reaction induced by EVs. Based on the performed experiment it is not clear whether the increased number of immune cells in the tumor tissue is due to enhanced tumor infiltration or a higher number of circulating NK, T and B cells.
  4. Did the authors observe a difference in the activation level of CD4 T cells? Figure 1 shows an upregulation of CD80, CD86 and MHC-II on DCs which may indicate an involvement of CD4 T cells. To further study this possibility, a CD4 T cell depletion experiment would be informative.
  5. What is the mechanism underlying the described findings? Are EVs taken up by DCs which then prime T cells? Do EVs directly interact with T cells? An experiment using FTY720 should be considered to answer this question?
  6. Line 159: Why would EVs enhance tumor infiltration? Based on the presented data it is not possible to distinguish between enhanced infiltration and an overall increase of CD8 T cells which would also result in an increase T cells in the tumor tissue. An experiment characterizing immune cell subsets following EVs administration (before tumor inoculation) would improve the interpretation and discussion of the findings shown in Figure 3 and 5.
  7. Line 172: This experiment demonstrates that the observed effect is dependent on CD8 T cells but does not proof antigen specificity.
  8. Line 200: EV administration results in an activation of immune cells (T cells, DCs, etc). Although not specific, that enhanced activation is expected to affect the growth of unrelated tumors.
  9. The authors mention that EV priming is not sufficient to induce an anti-tumor effect in mice with established tumors and interpret the lacking immune response as a result of MHC downregulation (without supporting data). However, cellular therapies are also challenged by several other solid tumor characteristics (immunosuppressive TME, lacking tumor infiltration, antigen loss, etc.). The authors should further expand this section and highlight the crucial timing of EVs to ensure their anti-tumor function
  10. For translational purposes, is it possible to isolate BC EVs from tumor-bearing mice and use those EVs to prime tumor-free mice before tumor inoculation?

MINOR COMMENTS

  1. Supplementary figures should be mentioned/discussed in the result section
  2. Are data representative for one experiment with technical replicates? Or pooled data from several experiments?
  3. In addition to CD69, serum cytokines could be analyzed to further study T cell activation
  4. The authors should consider performing (or discussing) TCR sequencing to further characterize the CD8 T cell response and to strengthen the tumor antigen-specificity statement.
  5. Line 165: “… it was later confirmed confirmed by…” needs rewording
  6. Line 176: “… demonstrating the average size tumor size…” needs rewording.
  7. Line 292: How can cytokines in the administered EVs recruit immune cells to tumor tissue? Wouldn’t the cytokines only affect DCs and possibly T cell priming? The association with tumor recruitment is not clear.
  8. Why would EVs prevent cancer reoccurrence in patients? The primary tumor would already activate the immune system resulting in memory T cells. Would EVs function as a booster without inducing a novel immune response? If the primary tumor does not induce an immune response, it’s unlikely that EVs could be used as priming agent to induce an immune response.

Author Response

MAJOR COMMENTS/CHANGES

  1. Quality of the figures should be improved (axis labeling and scaling are blurry).

Answer: We have improved the quality of each figure, and the resolution of all revised figures have now achieved 300 dpi.

  1. Figure 2D: Why is the percentage of tumor cells in the tumor tissue so low (even in the PBS group)?

Answer: After carefully reviewing all data, we recognized that the data presented in the old figure 2D was derived from the original trial. Since the first trial, we had performed a few biological repeats and optimized the experimental conditions with proper gating strategies (consulted with the co-author Dr. Gerber, an immunologist, who designed this multi-channel flow cytometry analysis used in this study). We now are confident to report an average of 30% tumor cells in tumor samples from the PBS control vs.  much fewer tumor cells in tumors from the EV-priming group. Therefore, in this revised manuscript, we replaced the old figure 2D with a representative results obtained from one of the biological repeats; it shows higher percentage of tumor cells in both groups, but the conclusions and statistical differences are the same. Please see the revised Figure 2D. We apologize for our oversight. 

  1. Figure 3: In addition to tumor tissue, spleen, blood and/or lymph nodes should be analyzed to further characterize the immune reaction induced by EVs. Based on the performed experiment it is not clear whether the increased number of immune cells in the tumor tissue is due to enhanced tumor infiltration or a higher number of circulating NK, T and B cells.

Answer: Even though we agree with reviewer that analyzing other tissues, such as blood, lymph nodes and spleen, might add additional mechanistic details of the mode of EV action and add to the complexity of EV-mediated immune networking which is beyond the scope of this manuscript. We appreciate the suggestion and have included it as a future prospect in the revised manuscript.  Please find the modifications in line 376 to highlight the importance of elucidating this possibility.

  1. Did the authors observe a difference in the activation level of CD4 T cells? Figure 1 shows an upregulation of CD80, CD86 and MHC-II on DCs which may indicate an involvement of CD4 T cells. To further study this possibility, a CD4 T cell depletion experiment would be informative.

Answer: This is a great idea that we could have pursued. However, we did not see statistically altered CD4 T cells numbers in the EV-primed tumors (day 11 and day 19) as shown in Fig 3B, so we did not further dissect CD4 T cells’ involvement in EV anti-tumor action. Instead, we found a significant change in infiltrated CD8 T cells in the EV-primed tumors, and decided to focus on proving EV’s ability to mediate the immune response through CD8 T cells via depleting these lymphocytes in vivo.

  1. What is the mechanism underlying the described findings? Are EVs taken up by DCs which then prime T cells? Do EVs directly interact with T cells? An experiment using FTY720 should be considered to answer this question?

Answer: Thanks for the suggestion. In this revised manuscript, we have included statement to summarize the underlying mechanism based on our data.  “The underlying mechanism of such action starts with the release of immunogenic EVs by tumor cells, and those TEVs were internalized and processed by DCs. The EV-activated DCs would activate tumor-suppressive immune cells, including but not limited to CD8+ T cells, then provoke an anti-tumor immune response.” (please see the line 440 in the Discussion section). However, it is still possible that EVs directly interact with T cells in the draining lymph nodes of the priming injection site.

  1. Line 159: Why would EVs enhance tumor infiltration? Based on the presented data it is not possible to distinguish between enhanced infiltration and an overall increase of CD8 T cells which would also result in an increase T cells in the tumor tissue. An experiment characterizing immune cell subsets following EVs administration (before tumor inoculation) would improve the interpretation and discussion of the findings shown in Figure 3 and 5.

Answer: Thanks for the comment. The increasing T cell infiltration into EV-primed tumors shown in our data could be the results of activation of T cells directly and indirectly by EVs. For instance, EVs can directly promote T cell proliferation, migration and/or indirectly function as chemoattractant for T cell infiltration. As suggested by reviewer, we have re-worded in the Result section (lines 207, 209, 210) added the possible mechanisms and the limitation of the current model in the Discussion section (line 370-376).

  1. Line 172: This experiment demonstrates that the observed effect is dependent on CD8 T cells but does not proof antigen specificity.

Answer: This is correct, our data we did not directly prove the antigen specific action of MB49EV against MB49 tumor growth, unless we can identify MB49 tumor antigen(s). However, the data from the colon 38 tumor model suggest that the anti-MB49 cell growth by MB49 EVs is dependent on CD8+ T lymphocytes, and partially via an antigen specific action.  We have re-worded, please see line 225 in the Result Section.

  1. Line 200: EV administration results in an activation of immune cells (T cells, DCs, etc). Although not specific, that enhanced activation is expected to affect the growth of unrelated tumors.

Answer: This comment is kindly appreciated. We have re-worded it in this revised manuscript, specifically in lines 248, and 260-263 in results section and 399-401 in the Discussion section. 

  1. The authors mention that EV priming is not sufficient to induce an anti-tumor effect in mice with established tumors and interpret the lacking immune response as a result of MHC downregulation (without supporting data). However, cellular therapies are also challenged by several other solid tumor characteristics (immunosuppressive TME, lacking tumor infiltration, antigen loss, etc.). The authors should further expand this section and highlight the crucial timing of EVs to ensure their anti-tumor function

Answer: This comment is very much appreciated. As suggested, we have expanded this section in the revised manuscript to include other potential hypotheses that could support our findings, please see line 348-356 in Discussion section. As indicated in this revised manuscript, we included this data in a manuscript to be submitted (Ortiz Bonilla, et, al, manuscript in preparation, 2022), thus, it is not included in this current manuscript.  

  1. For translational purposes, is it possible to isolate BC EVs from tumor-bearing mice and use those EVs to prime tumor-free mice before tumor inoculation?

Answer: This is a great idea, but technically challenging. First, the blood-derived EVs are heterogeneous, as they contain a mixed population of EVs derived from various cell types, including tumors, normal cells, stroma cells, etc. Unfortunately, tumor-specific EV marker(s) for sorting tumor specific EVs have not yet identified.  Thus, when priming EVs purified from blood of tumor-bearing mice, the impact of tumor-derived EVs might be overshadowed by EVs from other cell types. Second, the cancer cells within the tumors grown in mice might be different from the cancer cells cultured in vitro, which very likely influence their EV cargo contents. Thus, those TEVs collected from tumor-bearing mice, if isolated from the pool of EVs in circulation, might not have the same anti-tumor immune properties. However, it will be very interesting to test those ideas when we are able to isolate tumor specific blood EVs from tumor-bearing mice.

MINOR COMMENTS

  1. Supplementary figures should be mentioned/discussed in the result section

Answer: We agree, and have added them within the text (please see line 105, 266 and 269 in the Results section).

  1. Are data representative for one experiment with technical replicates? Or pooled data from several experiments?

Answer: Selected data/graphs/plots are representative of multiple biological repeats; each one contained multiple technical replicates.

  1. In addition to CD69, serum cytokines could be analyzed to further study T cell activation

Answer: This is a good suggestion. Studying the systemic immune response provoked by TEVs, including free cytokines and many other factors in the serum, would be informative. However, it is explorative in nature and not within the scope of current study.  

  1. The authors should consider performing (or discussing) TCR sequencing to further characterize the CD8 T cell response and to strengthen the tumor antigen-specificity statement.

Answer: Thank for pointing out this important aspect. We have discussed its importance within the Discussion session.

  1. Line 165: “… it was later confirmed confirmed by…” needs rewording

Answer: It was grammatically corrected.

  1. Line 176: “… demonstrating the average size tumor size…” needs rewording.

Answer: It was grammatically corrected.

  1. Line 292: How can cytokines in the administered EVs recruit immune cells to tumor tissue? Wouldn’t the cytokines only affect DCs and possibly T cell priming? The association with tumor recruitment is not clear.

Answer: We acknowledge the explorative nature of cytokine data, but also value the potential implications of EV encapsulated cytokines in mediating immune response. Therefore, we have revised our data interpretation as a possible mechanism that requires further confirmation (please see line 260-263 in the Results section and 399-402 in the Discussion section). 

  1. Why would EVs prevent cancer reoccurrence in patients? The primary tumor would already activate the immune system resulting in memory T cells. Would EVs function as a booster without inducing a novel immune response? If the primary tumor does not induce an immune response, it’s unlikely that EVs could be used as priming agent to induce an immune response.

Answer: This is a valid point, and a possible clinical scenario. Those TEVs from the primary tumors are very likely not to be able to induce sufficient anti-tumor immunity under the influence of an immuno-suppressive tumor microenvironment. However, the main message conveyed from our study is that we presented evidence of tumor preventive action by TEVs, and provided an experimental model allowing the discovery of the key molecule(s) and their downstream pathways that might provide a novel therapeutic strategies to prevent tumor recurrence.       

Reviewer 2 Report

This study showed TEVs augmented anti-tumor immunity in MB49 tumor-bearing mice models. I think several experiments should be added prior to acceptance.

1) How do the the priming of TEVs affect immune cells in spleen and lymph nodes in vivo? Do TEVs activate dendritic cells?

2) In Figure 3C, the activation of CD8 lymphocytes were evaluated by the status of CD69. I think CD69 reflects immediate activation of anti-tumor immunity. In addition to CD69, I think it is better to assess CD25 or IFNγ status of CD8 lymphocytes.  

3) In Figure4B and 4D, multiple comparison is necessary to assess statistical significance.

4) Do the priming by TEVs prevent the development of metastasis? The intravenous injection of tumor cells after priming enables such an evaluation.

Author Response

Comments:

  • How do the priming of TEVs affect immune cells in spleen and lymph nodes in vivo? Do TEVs activate dendritic cells?

Answer: These are interesting questions, and similar concern raised by reviewer 1. Please refer to our answer to comment #3, by reviewer 1.

  • In Figure 3C, the activation of CD8 lymphocytes were evaluated by the status of CD69. I think CD69 reflects immediate activation of anti-tumor immunity. In addition to CD69, I think it is better to assess CD25 or IFNγ status of CD8 lymphocytes. 

Answer: We agree that other markers could have been added to the flow cytometry antibody panel to better understand T cell activation. However, many researchers in our institute share the panel design, so we were not able to include other markers at this moment.  Even though CD69 is commonly known as an early marker for cytotoxic lymphocytes activation, there is evidence showing consistent expression of CD69 as an activation marker for CD8 T cells, specifically in experiments where multiple and continuous boostings/primings are performed (Dorta-Estremera, 2018, IJRDBP). Therefore, we are confident that the observed readouts represent CD8 T cell activation and recognize that other markers will bring a broader understanding of their activation status, as considered in the Discussion section.

  • In Figure 4B and 4D, multiple comparison is necessary to assess statistical significance.

Answer: Thank you for pointing it out. We have re-run our statistical analysis, as indicated in each figure legend and in the Methods and Materials section (statistical analysis).

  • Do the priming by TEVs prevent the development of metastasis? The intravenous injection of tumor cells after priming enables such an evaluation.

Answer: This is a very interesting question. It is well documented that TEVs play major roles in pro-tumorigenic processes; however, TEVs also contain many cargo molecules with opposite roles. We are still at the early stage of understanding the dynamic and heterogeneous nature of TEVs during cancer initiation, progression, and metastasis, as well as complexity of TEV-mediated communication among various cell types within the tumors and TME. We are encouraged by our findings, but also acknowledge that more research needs to be done, such as the reviewer’s suggestion, to better understand EVs’ role in cancer biology. 

Round 2

Reviewer 2 Report

The authors answered my questions correctly. But I think one more revision is necessary. In conclusion, the authors commented "this work provides a valuable platform that will guide future research efforts to better understand the potential of BC EVs as a patient-specific therapeutic cancer vaccine or adjuvant therapy to reduce the incidence of tumor recurrence". This study did not show the prevention of recurrence by TEVs. So, the comment of adjuvant therapy should be excluded or the potential of TEVs as an adjuvant therapy should be evaluated. It could by tested by the administration of TEVs after the resection of the inoculated tumor, followed by reinoculation of the second tumor.

Author Response

To be more conservative, we removed the statement “adjuvant therapy for prevention of tumor recurrence” (line 447) from the Discussion section as well as reworded similar statements in the Abstract and Result sections, according to reviewer’s suggestion